# Novel Brønsted Acidic Ionic Liquids as High Efficiency Catalysts for Liquid-Phase Beckmann Rearrangement

Chunxiao Ren [1,2,†], Zhiyuan Wang [1,†], Qingwei Gao [1], Jisheng Li [1], Siqi Jiang [3], Qizhong Huang [1], Ye Yang [1], Jiahui Zhang [1], Yajing Wang [1], Yufeng Hu [1,*], Zhichang Liu [1,*] and Xin Guo [1,*]

[1] State Key Laboratory of Heavy Oil Processing and High Pressure Fluid Phase Behavior and Property Research Laboratory, China University of Petroleum, Beijing 102249, China; renchunxiao@petrochina.com.cn (C.R.); wangzy9737@163.com (Z.W.); 2020215342@student.cup.edu.cn (Q.G.); lijisheng@isl.ac.cn (J.L.); qizhong.huang@chnenergy.com.cn (Q.H.); yyx2205@sina.com (Y.Y.); zjh15544039035@163.com (J.Z.); wyj15833871259@126.com (Y.W.)

[2] Petrochina Petrochemical Research Institute, Beijing 102206, China

[3] Sinopec Engineering Incorporation, Engineering Technology R&D Department, Beijing 100101, China; jiangsiqi@sei.com.cn

\* Correspondence: huyf3581@sina.com (Y.H.); lzch@cup.edu.cn (Z.L.); guoxin19940620@163.com (X.G.)

† These authors contributed equally to this work.

**Abstract:** Exploring environmentally friendly, efficient, cheap and recyclable catalysts are essential for the development of green, sustainable and mild processes for the liquid-phase Beckmann rearrangement. Herein, a novel caprolactam-based Brønsted acidic ionic liquid ([CPL][2MSA]) was developed for the conversion of cyclohexanone oxime (CHO) to caprolactam (CPL), not only as a catalyst, but also as a mild reaction medium. Under the reaction conditions for the reaction temperature (90 °C), reaction time (2 h) and mole ratio ([CPL][2MSA]: CHO = 3:1), [CPL][2MSA] possesses plenty of high sulfonate groups, which exhibit high conversion (100%) and selectivity (95%) without any other co-catalysts or metals. Based on the thermogravimetric (TGA) and differential scanning calorimetry (DSC) analyses, the decomposition and glass transition temperatures are gradually increased with the increase in MSA mole content, revealing the existence of hydrogen-bonded clusters. Interestingly, the occurrent route of the liquid-phase Beckmann rearrangement for CHO in [CPL][2MSA] is revealed by in situ FT-Raman. In addition, the dominating H-bond combination between CHO and [CPL][2MSA] is further confirmed by COSMO-RS model. The activation energy ($E$a) of the reaction is calculated by the first-order reaction kinetics. Thus, the [CPL][2MSA] with plenty of acidic catalytic active species is an environmentally friendly and efficient candidate for the liquid-phase Beckmann rearrangement.

**Keywords:** Brønsted acidic ionic liquid; Beckmann rearrangement; kinetics; H-bond

## 1. Introduction

The transformation of cyclohexanone oxime (CHO) to caprolactam (CPL), known as the Beckmann rearrangement (Scheme 1), has been researched in many fields, and is widely used in pharmaceuticals, textile and the electronic industries. CPL, as a precursor, is used to prepare the detergents, lubricants, Nylon-6 and resins [1–4]. In the liquid-phase Beckmann rearrangement, sulfuric acid and oleum, as catalysts and reaction medium, have been used for the industrial processes. One crucial issue is that sulfuric acid and oleum as catalysts produce a large amount of ammonium sulfate (as by-product) and cause serious erosion of the equipment. In addition, the neutralization of these acids is unavoidable when using ammonium hydroxide, and the formation of ammonium sulfate prevents the adequate recovery of CPL by incurring a 10–15% product loss [5–7]. At present, a method of vapor-phase Beckmann rearrangement is attracting researchers. In the vapor-phase rearrangement of CHO, a variety of solid acid catalysts have been investigated to replace the environmentally unfriendly process using sulfuric acid, including zeolites,

heteropoly acids (HPAs), sulfonated carbon materials, ion exchange resins and supported metal oxides [8–12]. However, these above solid acid catalysts also have disadvantages, such as catalyst deposition and deactivation. It is a key point that the vapor-phase process always occurs at a highly reactive temperatures ($T > 300 \, °C$), causing high energy consumption, and often leads to low selectivity and fast catalyst deactivation arising from coke formation. In addition, it is difficult for the Beckmann rearrangement process to change some important oximes to lactams, due to the instability of the precursors in the vapor-phase [13,14]. Therefore, the development of a mild and green liquid-phase catalytic Beckmann rearrangement process is urgently required.

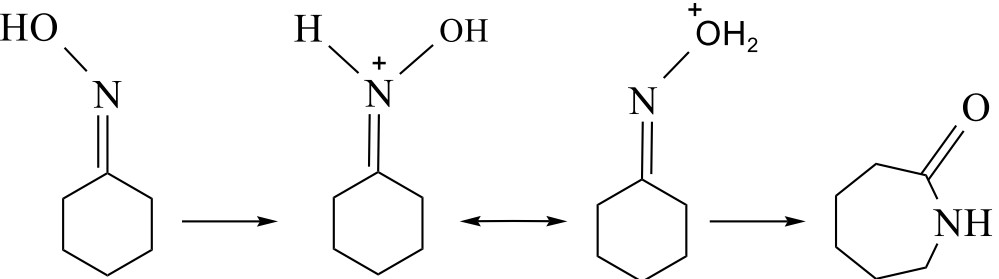

**Scheme 1.** Beckmann rearrangement of cyclohexanone oxime to caprolactam.

In the liquid-phase Beckmann rearrangement, some reports have been made following those of [15–20]. Srivastava et al. [21], reported a catalytic system in which [$C_3SO_3Hmim$][TFA] (catalyst) and $ZnCl_2$ (additives) achieved a high selectivity (99%) for the Beckmann rearrangement. Wang et al. [22] completed the first study of an in situ incorporated sulfonic group and polymeric skeleton (H-PDVB-$SO_3H$) for use as catalyst in the liquid-phase Beckmann rearrangement; H-PDVB-$SO_3H$ exhibited a high yield (75%) and excellent repeatability. Xia et al. [23] synthesized a series of novel Brønsted acidic ionic liquids (ILs), and a high yield was obtained by using ILs–$ZnCl_2$ catalysts. However, the synthesis processes for the above-mentioned catalysts are complex and the price of raw materials is expensive industrially. Therefore, it is particularly important to develop a novel liquid-phase Beckmann rearrangement catalyst in order to overcome these problems.

The caprolactam-based Brønsted acidic ILs have been reported as novel catalysts and reaction media for the Beckmann rearrangement of CHO to CPL. Caprolactam-based Brønsted acidic ILs have more advantages than imidazolium, pyridine and quaternary ammonium, such as cheapness, environmental benignity, easy preparation, etc. Furthermore, a great challenge is that organic solvents (acetonitrile, benzonitrile, dimethyl formamide) as the reaction medium can lead to higher costs and to the environmental pollution associated with sustainability concerns [24]. Unfortunately, the existence of the above problems may hinder the development of CPL industry. Therefore, the road towards a CPL industry involves much work to be done in achieving green chemistry processes.

Recently, Brønsted acidic ILs with $SO_3H$ groups have been used in the catalytic reaction, including 3-methyl-1-(butyl-4-sulfonyl) imidazolium hydrogen sulfate ([$HO_3SBMIM$][$HSO_4$]), hydroxylamine *N,N,N*-trimethyl-*N*-sulfobutyl hydrosulfate salt ([$NH_2OH$]$_2$[$HSO_3$-b-*N*($CH_3$)$_3$][$HSO_4$]), [3-(1-methylimidazolium-3-yl)propane-1-sulfonate]$_3$ $PW_{12}O_{40}$([$MIMPS$]$_3PW_{12}O_{40}$) [25–27]. Their superior advantages are that Brønsted acidic ILs with $SO_3H$ groups, for example 3-methyl-1-(butyl-4-sulfonyl) imidazolium tosilate ([MBsIm][Ts-OH]), 3-hexyl-1-(butyl-4-sulfonyl) imidazolium trifluoromethanesulfonate ([HBsIm][OTf]), ect. have provided satisfactory acid strength to catalyze the Beckmann rearrangement of several oximes [23]. Interestingly, some researches in recent decades have investigate functionalized ILs with $SO_3H$ groups, used as the reaction catalyst for the Beckmann rearrangement of CHO into CPL. However, in some cases, various amounts of the $SO_3H$ groups need to be introduced into ILs in order to enhance acidity, and thus to improve the catalytic effect [28,29].

In this paper, a novel caprolactam-based Brønsted acidic ionic liquid ([CPL][2MSA]) is directly synthesized through one-step solvent-free based on the buffer solution theory. The catalytic performances are evaluated in the liquid-phase Beckmann rearrangement. When the reaction time is 2 h, [CPL][2MSA] shows a high conversion (100%) and selectivity (95%) without other any co-catalysts or metals. On the bases of the thermogravimetric analysis (TGA) and differential scanning calorimetry (DSC), with an increase in MSA mole content, the obtained results show that the decomposition and glass transition temperatures are gradually increased. According to in situ FT-Raman, it revealed the occurrent route for the liquid-phase Beckmann rearrangement of CHO into CPL. Interestingly, the results of the COSMO-RS calculation show that the transformation of CHO into CPL is mainly attributed to the intermolecular interaction (H-bond). On the basis of the kinetic model calculation results, the first order reaction kinetics satisfy the conversion process of CHO to CPL, and activation energy ($Ea$) of the reaction is calculated as 29.8 KJ mol$^{-1}$. More importantly, CPL, as one component of ILs used, can be conjectured to achieve a dynamic exchange between CPL-based ILs and the produced CPL during the rearrangement reaction, and this would be largely avoided its strong chemical combination with acidic catalyst. In addition, [CPL][2MSA] with the existence of abundant acidic catalytic active species, as a novel catalyst, shows excellently selective for the liquid phase Beckmann rearrangement.

## 2. Results and Discussion

*Characterization of the Synthesized [CPL][XMSA]*

The $^1$H NMR and $^{13}$C NMR of the synthesized [CPL][2MSA] are shown in Figure S1a,b. [CPL][2MSA] exhibits an excellent structure. The structure of [CPL][*XMSA*] is further confirmed by FT-IR spectra. From Figure 1a, it can be seen that the single bond of S–O stretching vibration appears in range of 840 cm$^{-1}$–945 cm$^{-1}$, the double bond of S=O stretching vibration is listed in 1000 cm$^{-1}$–1150 cm$^{-1}$ and the bond of S–OH stretching vibration fall from 1340 cm$^{-1}$ to 1400 cm$^{-1}$ [28,30]. For [CPL][MSA], it only contains S–OH vibration (842.2 cm$^{-1}$), S=O stretch (1020.5 cm$^{-1}$) and a complex multiple vibration of SO$_3$ unit (the most intense bond at 1150.5 cm$^{-1}$). With the increase of MSA molar ratios, the structure of [CPL][2MSA] or [CPL][3MSA] is similar to that of MSA, and the S–OH vibration occurs blue shift from 842.2 cm$^{-1}$ to 942.5 cm$^{-1}$. In addition, the S = O stretching vibration ([CPL][2MSA], [CPL][3MSA]) is intermediate between MSA and [CPL][MSA], since this motion is restricted by very strong hydrogen bonding in the cluster. Based on the UV–Vis DRS spectra (Figure 1b), the Hammett Brønsted acid scales ($H_0$) of [CPL][*XMSA*] are tested. The $H_0$ values of [CPL][MSA], [CPL][2MSA], [CPL][3MSA] are 2.43, 2.03 and 1.89 in Table 1 (Entry 2−4), respectively. The results show $H_0$ gradually becomes stronger with increase of MSA content. In order to confirm the acid values, 0.05 mol/L KOH solution is used to titrate the [CPL][*XMSA*] (Table 1) by using phenolphthalein as the indicator, and the result is consistent with $H_0$. According to thermogravimetric analysis (TGA), the synthesized CPL in different molar ratios (MSA) were tested in Figure 2a. With increase of MSA molar ratio, the decomposition temperatures ($T_g$) increase slightly, indicating the formation of hydrogen bonds between CPL and MSA. Compare with CPL, $T_g$ of synthesized [CPL][MSA] decreases with the increase of MSA content, indicating that more hydrogen bonds (H-band) are formed between CPL and MSA. Due to the presence of the strong H-bond, the structure of [CPL][*XMSA*] is really stable. As shown in Figure 2b, the glass transition temperatures ($T_g$) of CPL and [CPL][*XMSA*] are investigated in the range of –22–10 °C. The result shows that $T_g$ of [CPL][*XMSA*] are much lower than that of CPL. The result indicates that H-band between CPL and MSA decreases the melting point of CPL, ultimately forming a homogeneous and transparent liquid at room temperature. The detailed data of the TGA and DSC of the samples are listed in Table 2.

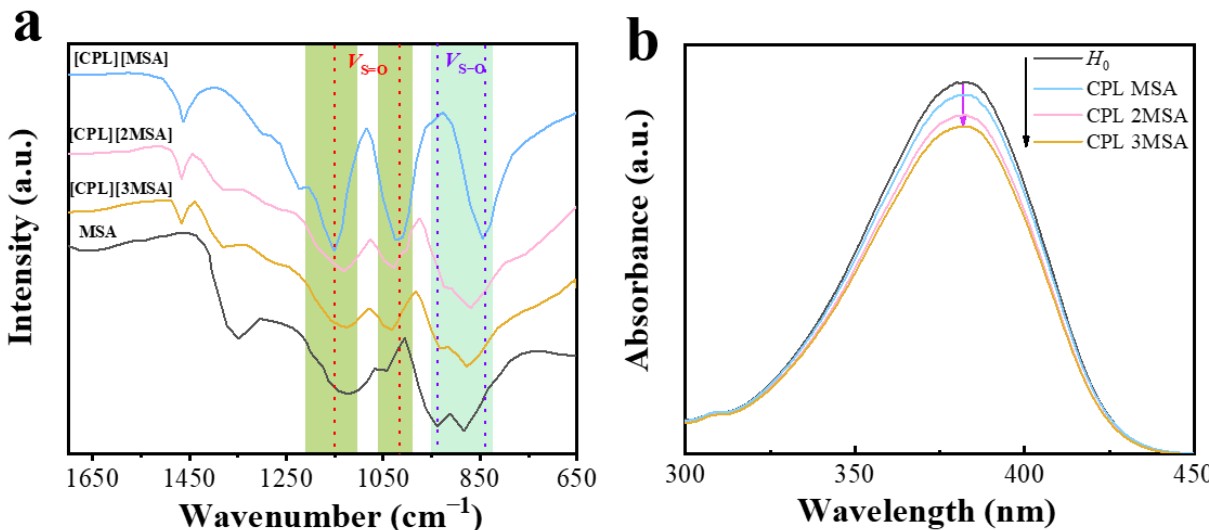

**Figure 1.** (**a**) FT-IR spectra and (**b**) UV–Vis DRS spectra of [CPL][MSA], [CPL][2MSA] and [CPL][3MSA].

**Table 1.** Acidity of [CPL][XMSA] synthesized with different molar ratios of MSA.

| Entry | ILs | $A_{max}$ | $H_0$ | Acid Value (0.05 mol KOH/mL) |
|---|---|---|---|---|
| 1 | Blank | 2.73 | — | — |
| 2 | [CPL][MSA] | 2.63 | 2.43 | 10.87 |
| 3 | [CPL][2MSA] | 2.48 | 2.03 | 15.39 |
| 4 | [CPL][3MSA] | 2.39 | 1.89 | 18.25 |

$(H_0 = pKa(I) + lg[I]_s/[HI^+]_s$[31], $A_{max}$ (Maximum absorbance)

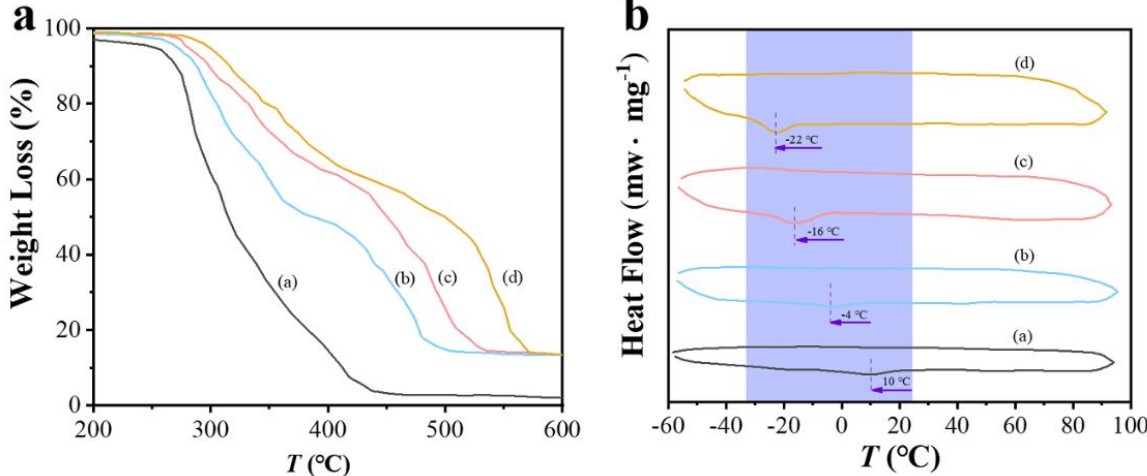

**Figure 2.** (**a**) TGA spectra of CPL (a), [CPL][3MSA] (b), [CPL][2MSA] (c) and [CPL][MSA] (d). (**b**) DSC spectra of CPL (a), [CPL][MSA] (b), [CPL][2MSA] (c) and [CPL][3MSA] (d).

Table 2 also lists the physicochemical properties of [CPL][XMSA], the conductivity of [CPL][MSA] ($6.67 \times 10^{-4}$ S/cm$^{-1}$) is slightly lower than that of [CPL][2MSA] ($4.45 \times 10^{-4}$ S/cm$^{-1}$) and [CPL][3MSA] ($3.24·10^{-4}$ S/cm$^{-1}$). As shown in Figure S2, relatively narrow electrochemical windows (1.7–2.1 V) were observed due to the presence of H-bond in [CPL][XMSA]. The result shows that the electrochemical windows also become wider with the increase of MSA content in CPL, which is consistent with the results of conductivities. The molecular orbitals represent the electron density in space, reflecting

chemical and physical properties. Further, the chemical and physical properties are evaluated by these functions in reported literature [32–36]. In this work, the frontier molecular orbital properties for [CPL][XMSA] are optimized through B3LYP/6-311G+(d,p) basis set. The stable conformation of structure and the spontaneity of any reaction can be predicted by thermodynamic properties, such as HOMO–LUMO gap ($\Delta E$), free energy and Dipole moment (Debye) [32,37]. In Table 3, [CPL][2MSA] exhibits the highest level of dipole moment, which will accelerate the chemical bond formation, non-bonding interaction and binding affinity. In addition, the free energy of all substances is negative, indicating that the reaction will occur spontaneously. Based on the values of HOMO and LUMO (Table 3), the $\Delta E$ are calculated according to $\Delta E$ = HOMO–LUMO gap. As shown in Figure 3, HOMO–LUMO gap ($\Delta E$) also decreases with the increase of MSA, suggesting that the introduction of MSA can improve the conductivity of CPL. In order to verify the reactive nature of any given compound, molecular electrostatic potentials of CPL, [CPL][MSA], [CPL][2MSA] and [CPL][3MSA] are calculated. With the increase of MSA, the molecular electrostatic potentials (MEP) of the compound changes accordingly, as shown in Figure 4. In addition, the density and viscosity of [CPL][XMSA] slightly increase with the introduction of MSA, indicating that [CPL][XMSA] has adjustable physical and chemical properties, which is more conducive to screening material suitable for rearrangement reactions.

**Table 2.** The physicochemical properties of CPL, [CPL][MSA], [CPL][2MSA], [CPL][3MSA].

| Ionic Liquid | $T_g$ (°C) | $T_d$ (°C) | Conductivity (S/cm$^{-1}$) | Density (g/cm$^3$) | Viscosity (cP) | $E$ (V) | H$_2$O [a] (ppm) | H$_2$O [b] (ppm) |
|---|---|---|---|---|---|---|---|---|
| CPL | 10 | 259.3 | – | 1.03 | – | 1.7 | 87 | – |
| [CPL][MSA] | −4 | 271.2 | $6.67 \cdot 10^{-4}$ | 1.18 | 78 | 1.8 | 187 | 186 |
| [CPL][2MSA] | −16 | 276.4 | $4.45 \cdot 10^{-4}$ | 1.27 | 54 | 2.0 | 245 | 379 |
| [CPL][3MSA] | −22 | 287.7 | $3.24 \cdot 10^{-4}$ | 1.33 | 37 | 2.1 | 319 | 567 |

[a]: the water content before reaction; [b]: the water content after reaction.

**Table 3.** Selected thermodynamic parameters of CPL, [CPL][MSA], [CPL][2MSA] and [CPL][3MSA].

| Ionic Liquid | $E$(V) | $\varepsilon_{LUMO}$ | $\varepsilon_{HUMO}$ | Free Energy (Hartree) | Dipole Moment (Debye) |
|---|---|---|---|---|---|
| CPL | 6.78 | −0.2314 | −7.0114 | −365.6195 | 4.4833 |
| [CPL][MSA] | 6.31 | −0.2709 | −6.5809 | −1029.6246 | 1.4170 |
| [CPL][2MSA] | 5.99 | −0.2914 | −6.2814 | −1694.2665 | 5.2633 |
| [CPL][3MSA] | 5.58 | −0.3183 | −5.8983 | −2358.6316 | 3.4694 |

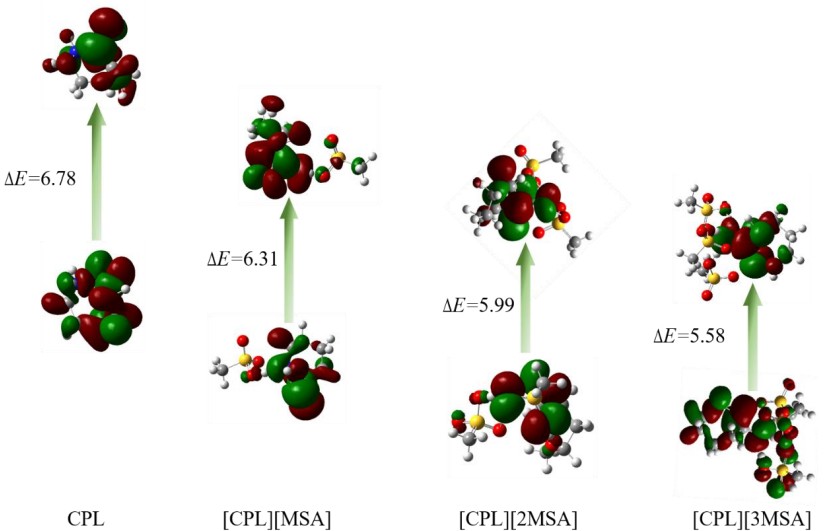

**Figure 3.** Frontier molecular orbital (HOMO–LUMO) and related transition energy of CPL, [CPL][MSA], [CPL][2MSA], [CPL][3MSA] and its degradants.

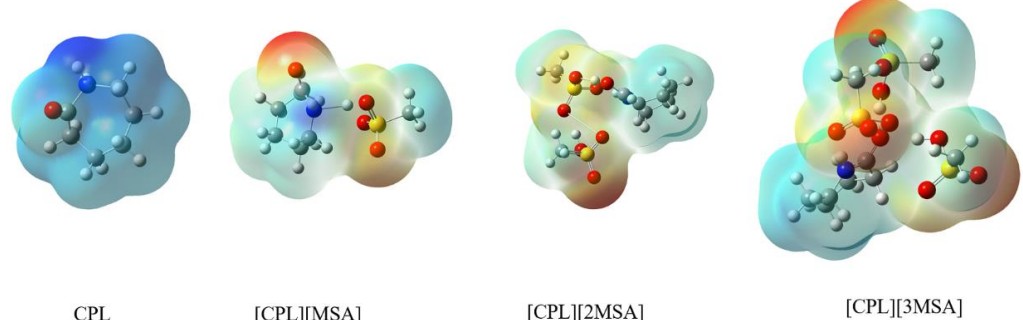

CPL       [CPL][MSA]       [CPL][2MSA]       [CPL][3MSA]

**Figure 4.** Molecular electrostatic potentials (MEP) of compounds (CPL, [CPL][MSA], [CPL][2MSA] and [CPL][3MSA]).

The catalytic activities of the synthesized [CPL][*X*MSA] for the rearrangement reaction are investigated and the results are summarized in Table 4. It can be seen that pure MSA exhibits low selectivity at 90 °C. Interestingly, the selectivity of [CPL][*X*MSA] (Entries 2–4) increases greatly from 47% to 94% at the same experimental conditions, and the experimental results show that the optimal catalyst is [CPL][2MSA]. For comparison, [CPL]Cl, [CPL][BF$_4$], [CPL][CSA], [CPL][2Cl], [CPL][2BF$_4$], [CPL][2CSA], [CPL][2HSO$_4$], [CPL][2TsOH] and [CPL][2TfOH] as catalysts and solvents were tested (Table 4, Entries 5–13) under the same experimental conditions. The experimental results show that the selectivity of the above-mentioned ILs catalysts was relatively low. According to experimental results, the important factor of Beckmann rearrangement reaction is acidity of ILs, but not the only factor. Interestingly, the Beckmann rearrangement may not be necessary to use a strong Brønsted acid as the catalyst and reaction medium. Figure 5a investigates the effects of different reaction temperatures on the experimental results. As the reaction temperature increases from 60 °C to 90 °C, the conversion of CHO and the selectivity of CPL gradually increase. When the reaction temperature is 100 °C, the conversion of CHO and the selectivity of CPL sharply decrease. In addition, Figure 5b also investigates the effect of different reaction times; the results show that the optimal reaction time is 2 h. Figure 5c shows the influence of the molar ratio of [CPL][2MSA]/oxime from 1/1 to 5/1, As the molar ratio ([CPL][2MSA]/oxime) increases, the selectivity first increases and then decreases, and the optimum molar ratio is 2:1 ([CPL][2MSA]/oxime). Figure 5d further investigates the repeatability of [CPL][2MSA]; only a little change happens during ten consecutive cycles of catalytic activity for liquid-phase Beckmann rearrangement. The results show that [CPL][2MSA] has excellent repeatability in the Beckmann rearrangement process. Based on these experimental results, the reason for the reduction of conversion and selectivity is that CHO may occurs polymerization reaction and CPL may undergo ring opening and polymerization reactions at high temperature. In addition, the content of water (Table 2, Entry 8) in [CPL][*X*MSA] is detected by the Karl Fischer method, and the water content in [CPL], [CPL][MSA], [CPL][2MSA] and [CPL][3MSA] are 87 ppm, 187 ppm, 245 ppm and 319 ppm, respectively. We also tested the content of water after the reaction (Table 2, Entry 9). In comparison with the dried ILs before the reaction, the water content of [CPL][2MSA] and [CPL][3MSA] get slightly higher with the increase of MSA content. But, the MSA content is too high, resulting in high acidity of the ILs and low selectivity of the catalytic system. Therefore, the optimal [CPL][2MSA] used may be able to sequester the water formed during the reaction and thus minimize hydrolysis as a competing reaction. Due to the small amount of water contained in IL, it further catalyzes the hydrolysis of CHO to produce by-products (cyclohexanone). According to the above possibility, the route of the Beckmann rearrangement of CHO catalyzed by acidic [CPL][2MSA] is shown in Scheme 2.

**Table 4.** Results of the liquid-phase Beckmann rearrangement of cyclohexanone oxime to caprolactam.

| Entry | Catalysts | $n$ (ILs: CHO) (mol) | $T$ (°C) | $t$ (h) | *Conv.* (%) | *Sel.* (%) |
|-------|-----------|----------------------|----------|---------|-------------|------------|
| 1 | MSA | 3:1 | 90 | 2 | 100 | 29 |
| 2 | [CPL][MSA] | 3:1 | 90 | 2 | 100 | 47 |
| 3 | [CPL][2MSA] | 3:1 | 90 | 2 | 100 | 95.4 |
| 4 | [CPL][3MSA] | 3:1 | 90 | 2 | 100 | 65 |
| 5 | [CPL]Cl | 3:1 | 90 | 2 | 100 | 43 |
| 6 | [CPL][BF$_4$] | 3:1 | 90 | 2 | 100 | 14 |
| 7 | [CPL][CSA] | 3:1 | 90 | 2 | 100 | 34 |
| 8 | [CPL][2Cl] | 3:1 | 90 | 2 | 100 | 47 |
| 9 | [CPL][2BF$_4$] | 3:1 | 90 | 2 | 100 | 57 |
| 10 | [CPL][2CSA] | 3:1 | 90 | 2 | 100 | 84 |
| 11 | [CPL][2HSO$_4$] | 3:1 | 90 | 2 | 100 | 37 |
| 12 | [CPL][2TsOH] | 3:1 | 90 | 2 | 100 | 71 |
| 13 | [CPL][2TfOH] | 3:1 | 90 | 2 | 100 | 64 |

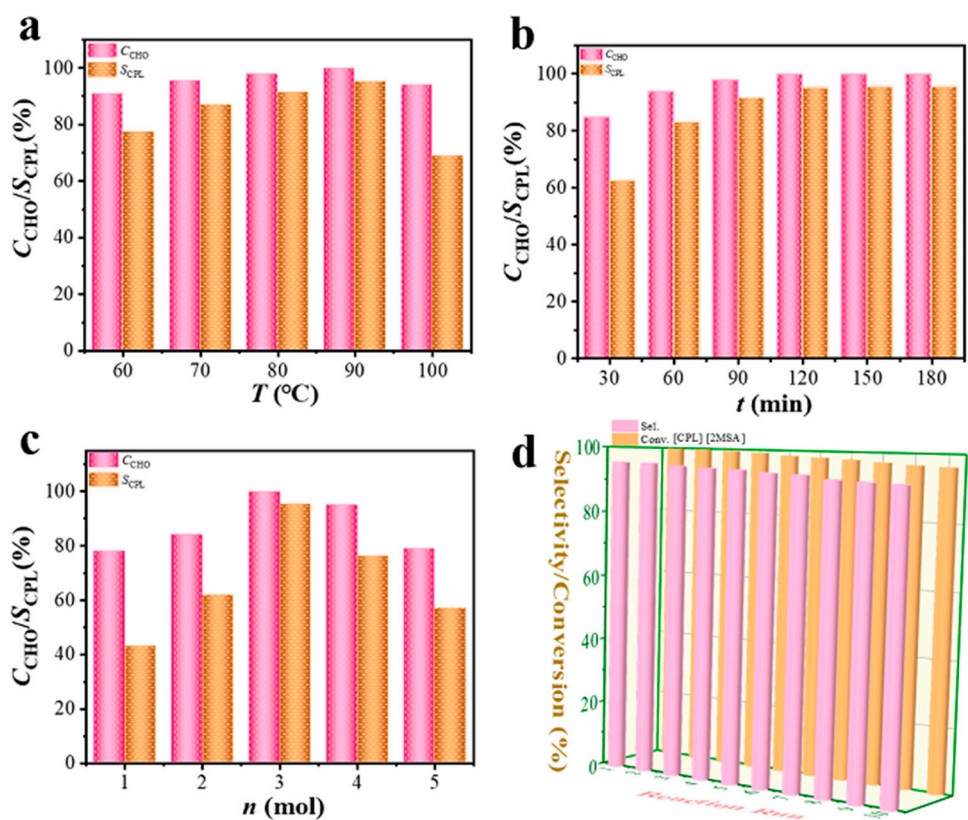

**Figure 5.** (**a**) The influence of reaction temperature, (**b**) reaction time, (**c**) molar ratio (**d**) catalytic reaction run of [CPL][2MSA] in the Beckmann rearrangement process.

COSMO-RS model as a powerful thermodynamic tool only requires the structural information to predict the solubility and other thermodynamic properties [38–43]. In our previous work, we successfully verified the accuracy of COSMO-RS in predicting the solubility of CHO in solutions against experimental results [44–46]. As shown in Figure S3, it predicts the solubility of CHO in [CPL][2MSA] at different temperatures; the result shows that the solubility of CHO gradually increases from 25 °C to 90 °C. However, when the temperature is above 90 °C, the solubility of CHO sharply descends, which is consistent with the experimental results. The $\sigma$-profile and $\sigma$-potential are two parameters in the COSMO-RS model. Then, the $\sigma$-potential represents the affinity of the system to a surface of polarity $\sigma$, $\sigma$-profile represents the ability of the substance to interact with itself or others, which were divided into three regions: (1) $\sigma < -0.0085$ e/Å$^2$: H-bond donor region;

(2) $\sigma > 0.0085$ e/Å$^2$: H-bond acceptor region; (3) $-0.0085$ e/Å$^2 < \sigma < 0.0085$ e/Å$^2$: the non-polar region. In order to verify the polarity and reactivity of substances, the $\sigma$-profiles and $\sigma$-potentials of CPL, MSA and CHO were calculated by using the COSMO-RS model as shown in Figure 6. From Figure 6a, the peaks of CPL, MSA and CHO are located in both the negative and positive regions, indicating that they can act as H-bond donor and acceptor. In addition, the peak intensity of MSA is higher than that of CPL and CHO due to the large number of oxygen atoms. In $\sigma$-potentials (Figure 6b), CHO only reflects the ability to interact with the H-bond donor. However, CPL and MSA not only exhibit the ability to interact with H-bond donor, but also with H-bond acceptor, which can better explain the favorable liquid phase Beckmann rearrangement ability of [CPL][2MSA], and this result is in accordance with the experimental results.

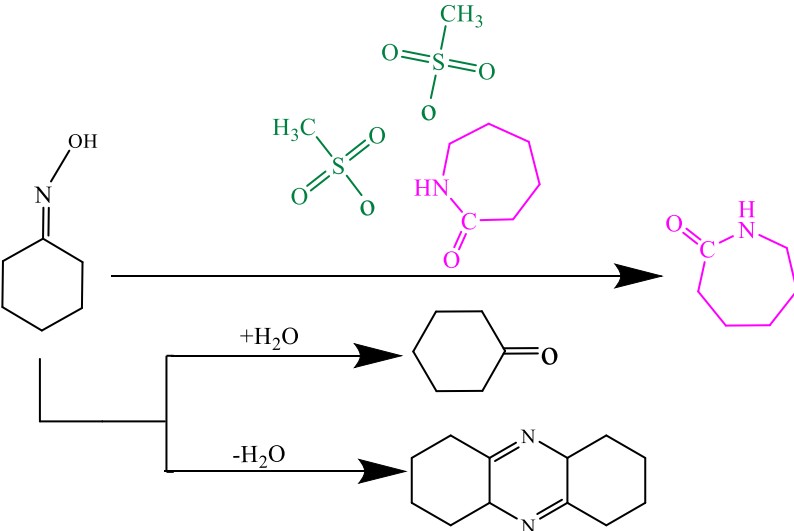

**Scheme 2.** The reaction path of Beckmann rearrangement of CHO in [CPL][2MSA].

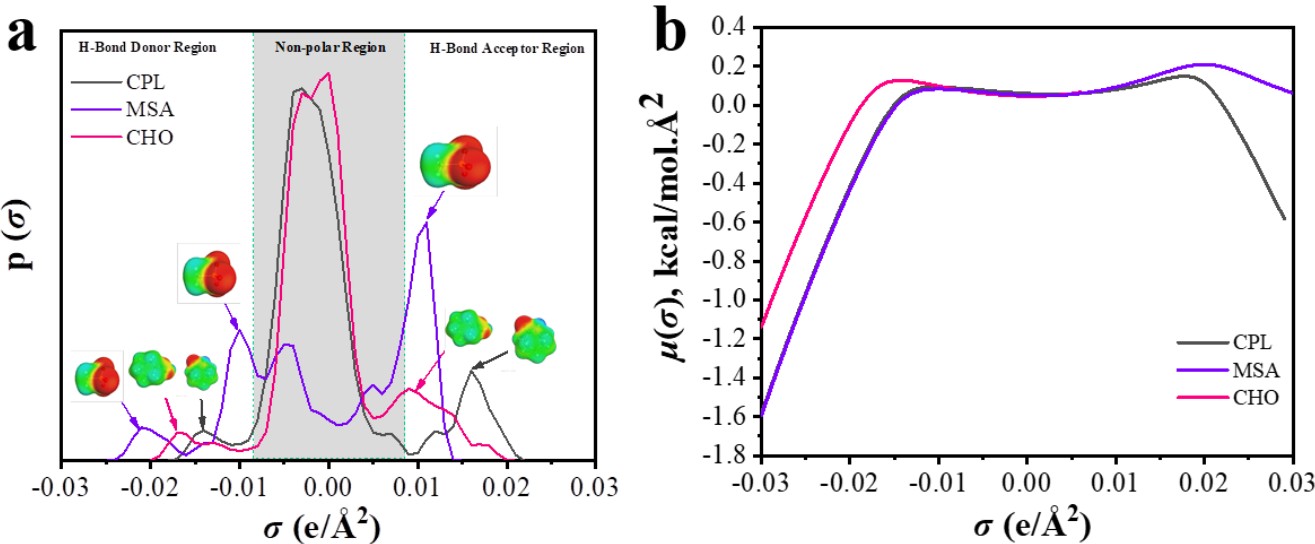

**Figure 6.** (**a**) $\sigma$-profiles and (**b**) $\sigma$-potentials of CHO, CPL, MSA.

In order to investigate the mechanism of the Beckmann rearrangement reaction of CHO, Figure 7 shows the in-situ FT-Raman spectra of CHO in [CPL][2MSA] under the different times, the typical C = N and C = O stretching vibration peak of pure CHO (a) and pure CPL (c) are located at 1660 cm$^{-1}$and 1637 cm$^{-1}$, respectively [24,47]. For [CPL][2MSA], the typical C = O stretching vibration peak is shifted from 1637 cm$^{-1}$ to 1690 cm$^{-1}$, indicating that the N of CPL may cause the protonation with MSA and further produce

H-bond between CPL and MSA. In addition, with the extension of reaction time, a new peak appears at 1673 cm$^{-1}$, indicating that the N of CHO is protonated on the surface of [CPL][2MSA] and then the peak strength gradually reduced. Meanwhile, the typical C = O stretching vibration peak of CPL becomes stronger, indicating the CHO transfer to CPL by the Beckmann rearrangement process. In addition, the excess enthalpies are also calculated by the COSMO-RS model of CHO in [CPL][2MSA]. The intermolecular interaction including misfit ($E_{MF}$), hydrogen bond interaction ($E_{HB}$) and van der Waals energy ($E_{vdW}$) are described in Equations (1)−(3). The excess enthalpy mainly consists of electrostatic misfit interaction energy (misfit), hydrogen bond (H−bond) interaction energy and van der Waals (vdW) interaction energy, which can be expressed by Equation (4) [40,48−50]. The detail calculation result of the excess enthalpy of the mixture is shown in Figure S4; it can be seen from the calculation result that the system is dominated by the H−bond. Based on the in-situ FT-Raman spectra and the calculation result, a possible catalytic mechanism for Beckmann rearrangement of CHO into CPL in [CPL][2MSA] has been proposed as shown in Scheme 3.

$$E_{MF}(\sigma, \sigma') = \alpha_{eff} \frac{\alpha'}{2} (\sigma, \sigma')^2 \tag{1}$$

$$E_{HB}(\sigma, \sigma') = \alpha_{eff} C_{HB} \min\left(0, \ \min(0, \sigma_{donor} + \sigma_{HB}) \max(0, \sigma_{acceptor} - \sigma_{HB})\right) \tag{2}$$

$$E_{vdW}(\sigma, \sigma') = \alpha_{eff}(\tau_{vdW} + \tau'_{vdW}) \tag{3}$$

$$E = E_{MF} + E_{HB} + E_{vdW} \tag{4}$$

where $\sigma$, $\sigma'$ are the screening charge densities of two different segments, $\alpha'$ is a general misfit constant, $\alpha_{eff}$ is the effective contact surface area, $C_{HB}$ and $\sigma_{HB}$ are the hydrogen bond coefficient and the cutoff of hydrogen bond, respectively. $\sigma_{donnor}$, $\sigma_{acceptor}$ represent the screening charge densities of hydrogen bond donor and acceptor segments, respectively; $\tau_{vdW}$ is the element-specific vdW interaction parameter.

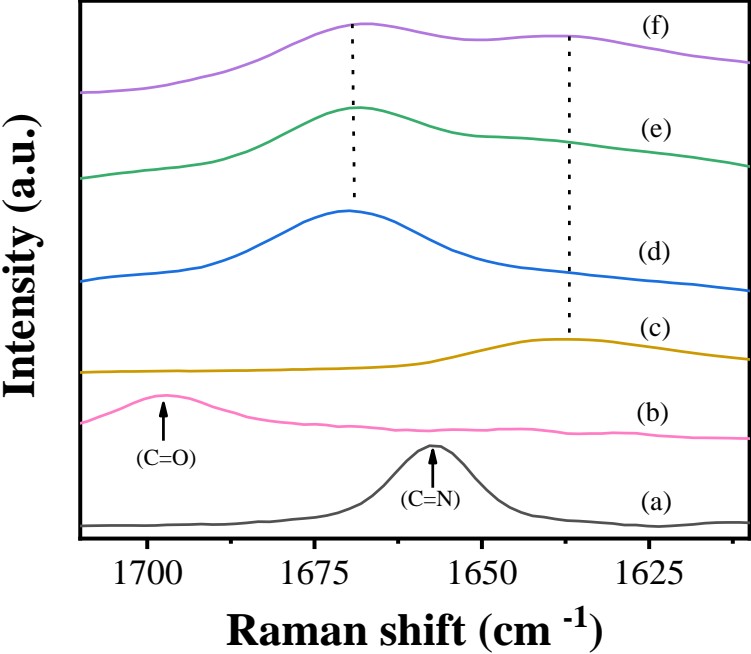

**Figure 7.** in-situ Raman spectra of Beckmann rearrangement of cyclohexanone oxime in [CPL][2MSA]. (a) CHO; (b) [CPL][2MSA]; (c) CPL; (d) 0 min (reaction mixture); (e) 60 min; (f) 120 min.

**Scheme 3.** A possible catalytic mechanism of the liquid-phase Beckmann rearrangement for CHO in [CPL][2MSA].

Based on the reaction mechanism, the kinetics of the CHO conversion to CPL are further studied in this work. Figure 8a shows the effect of CPL yield on different times without considering the influence of by-products, and the yield of CPL have no change within the 2 h. Therefore, it has been confirmed that a pseudo-first-order reaction is applied to the catalytic conversion of CHO to CPL by Equation (5) [6,51–53]. The yield of CPL significantly increases within the 2 h, indicating that high activation energy is required during this time. In addition, the reaction rate equation is expressed in terms of CPL yield by Equation (6). From Figure 8b, it can be seen that the rate constants with different times (30 min, 60 min, 90 min, 120 min) of the conversion of CHO to CPL in [CPL][2MSA] are 0.0101, 0.0148, 0.0186, 0.0252, respectively. According to Arrhenius equation, Figure 8c shows the Plot of $\ln k$ against $1/T$ and $Ea$ is calculated as 29.8 KJ mol$^{-1}$. In Table 5, the $Ea$ of a series of liquid catalysts are listed, it can be seen that this work is lower than other reported. Therefore, [CPL][2MSA] would be a promising catalyst for the liquid phase Beckmann rearrangement due to high yield, low activation energy and the mild reaction condition. In addition, we also explored the substrate scope of Beckmann rearrangement catalyzed by [CPL][2MSA], various ketoximes substrates are examined. In Table 6 (entries 1–4), aromatic ketoximes show the higher yields (92.7–98.4%) and cyclododecanone oxime also exhibits a comparative yield (94.9%) under the same experimental conditions. This again evidences the high catalytic activity of [CPL][2MSA] for Beckmann reactions.

$$\frac{d(C_{CHO})}{dt} = -kC_{CHO} \tag{5}$$

The reaction rate equation is expressed in terms of CPL yield by Equation (6):

$$\frac{d(Y_{CHO})}{dt} = k(1 - Y_{CPL}) \tag{6}$$

Equation (6) is integrated against reaction time to give Equation (7).

$$-\ln(1 - Y_{CPL}) = kt \tag{7}$$

where $k$, $t$, $C_{CHO}$ and $Y_{CPL}$ are the rate constant, the reaction time, the concentration of CHO and the yield of CPL, respectively.

$$\ln k = -\frac{Ea}{RT} + \ln A \tag{8}$$

where *A* is the pre-exponential factor, *R* is the gas constant (8.314 J (mol K)$^{-1}$) and *T* is the temperature in Kelvin; *E*a is the values of apparent activation energy.

**Table 5.** The Beckmann rearrangement over the recently reported catalysts.

| Catalysts | Experimental Conditions | *Ea*(kJ/mol) | Ref. |
|---|---|---|---|
| Trifluoroacetic acid (TFA) | 0.16 mol/L, TFA/acetonitrile = 2:1 | 93 (CHO), CPL (167) | [6] |
| Trifluoroacetic acid (TFA) | 0.16 mol/L, TFA/acetonitrile = 4:1 | 96 (CHO), CPL (159) | [6] |
| Trifluoroacetic acid (TFA) | 0.16 mol/L, TFA/acetonitrile = 9:1 | 92 (CHO), CPL (120) | [6] |
| Trifluoroacetic acid/oleum (T/A) | T/A ratio = 10, A/O ratio = 1.0 | 104.6 | [5] |
| Trifluoroacetic acid (TFA) | acetonitrile 15%wt., COX concentration 0.16 mol L$^{-1}$ | 169 | [18] |
| [$H_2SO_4$] + [$SO_3$] | ([$H_2SO_4$] + [$SO_3$])/([caprolactam] + [cyclohexanone oxime] = 1.4 | 254 | [54] |
| [CPL][2MSA] | [CPL][2MSA]: CHO = 2:1 | 29.8 | This work |

**Table 6.** Beckmann rearrangements of various substrates catalyzed by [CPL][2MSA].

| Entry | Oxime Substrate | Amide Product | *Yield* (%) |
|---|---|---|---|
| 1 | | | 96.2 |
| 2 | | | 98.4 |
| 3 | | | 92.7 |
| 4 | | | 94.9 |
| 5 | | | 95.1 |

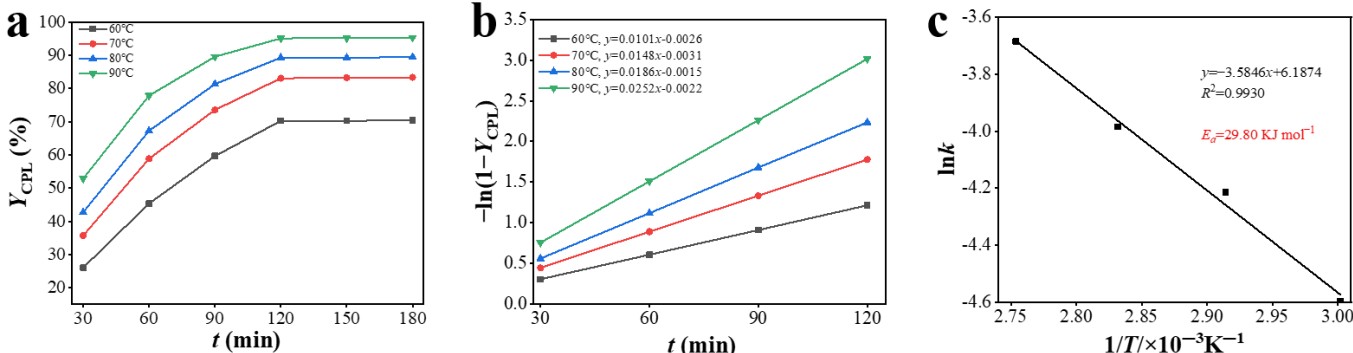

**Figure 8.** (**a**,**b**) The yield of CPL on the effect of different times; (**c**) Plot of In(*k*) against 1/T.

## 3. Materials and Methods

### 3.1. Chemicals

All reagents were purchased from Aladdin Industrial Corporation, (Shanghai, China) and used as received. Reagents: Methanesulfonic acid, *p*−toluenesulfonic acid (TsOH), trifluoromethanesulfonic acid (TfOH), hydrochloric acid (HCl), sulfuric acid ($H_2SO_4$), caprolactam, *n*-dodecane (GC 99.99 wt%), acetone.

### 3.2. Synthesis of ILs ([CPL][XMSA])

According to the reported method [55–57], a novel caprolactam-based Brønsted acidic ionic liquid ([CPL][2MSA]) was prepared through the one-step method as shown in Scheme 4. Firstly, methanesulfonic acid was added dropwise to a certain amount of caprolactam in an ice bath, and then the solution was stirred for 30 min and heated at 130 °C for 4 h. After reaction, a clear solution was extracted three times with acetone. Finally, the mixture solution was evaporated to remove extractant, and then the obtained product was dried in an oven at 80 °C for 12 h. The obtained sample was recorded as [CPL][*XMSA*], in which *X* stands for mole percent of MSA.[1]H NMR: δ 4.82 (s), 3.33 (s), 3.03 (m), 2.68 (s), 2.48 (s), 2.30 (m), 2.09 (s). [13]C NMR: δ 182.74, 42.64, 38.27, 33.70, 29.23, 27.10, 21.79.

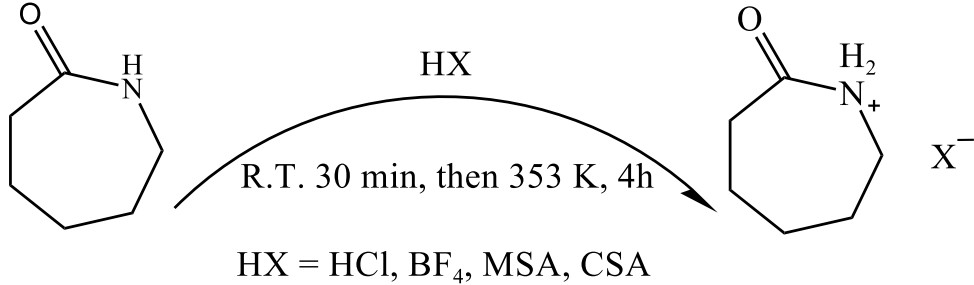

**Scheme 4.** The synthetic process of caprolactam-based Brønsted acidic ionic.

For comparison, [CPL][2Cl], [CPL][$2BF_4$], [CPL][2CSA], [CPL][$2HSO_4$], [CPL][2TsOH] and [CPL][2TfOH] were also fabricated with as–prepared [CPL][2MSA] through the same process.

### 3.3. Characterization

3.3.1. Nuclear Magnetic Resonance (NMR)

Nuclear magnetic resonance (NMR) ([1]H and [13]C) spectra were recorded by using the JEOL ECA−600 (JEOL, Tokyo, Japan) spectrometer at room temperature, with $CDCl_3$ as the solvent.

### 3.3.2. Fourier Transform Infrared (FT-IR) Spectrometer

FT–IR spectra of samples with KBr as the diluents were recorded on a Bruker Equinox 55 spectrometer (Bruker, Billerica, MA, USA).

### 3.3.3. In Situ FT-Raman

In situ Fourier transform (FT) Raman spectroscopy analysis was carried out in an in-situ diffuse reflectance pool with a Bruker Vector FT–IR spectrometer (6700) and MCT detector which was cooled by liquid $N_2$. Firstly, the [CPL][2MSA] and CHO were filled into the reaction cell with a molar ratio 3:1, and then the reaction cell was placed in the test chamber and heated to 90 °C under $N_2$ flow for 30 min to remove adsorbed impurities and the spectra were recorded in different times.

### 3.3.4. Thermogravimetric Analysis (TGA) and Differential Scanning Calorimetry (DSC)

TGA and DSC were carried out on a Q500 instrument and a Q2000 V24.11 Build 124, respectively. First, the sample was placed in a platinum pan with an average weight of 10–12 mg, then heated from 50 °C to 600 °C with a rate of 10 °C/min under a dry nitrogen atmosphere.

### *3.4. Catalytic Reaction*

[CPL][2MSA] and CHO (molar ratio of ILs/CHO = 1/1–3/1) were filled into a 25 mL three-necked round bottom flask under nitrogen atmosphere. Then, the mixture was stirred and heated to different temperatures (60, 70, 80, 90, 100, 110, 120) with a reflux condenser in an oil bath pan. After the reaction finished, 20 mL acetone was added to the three-necked round bottom flask, and then the product was separated by filtration and washed three times through organic solvent (acetone) from [CPL][2MSA]. Then, the organic solvent (acetone) was removed by using a rotary evaporator. Finally, the product was dried in an oven at 80 °C for 12 h. The products were characterized quantitatively with Kromat PC−17 column (30 m × 0.53 mm × 1.0 µm) SP3420 GC system equipped with FID detector and *n*-dodecane as the internal standard. According to the area of each chromatograph peak of all products, the concentrations were directly shown by the system of GC chemical station. The effect of ILs/CHO (molar ratio), reaction time, reaction temperature and the types of catalysts were explored according to the same experimental process (Section 3.4).

### *3.5. Catalyst Recycling*

In order to test the catalytic recyclability of [CPL][2MSA], the catalyst was extracted by organic solvent (acetone) three times after reaction and then dried in a vacuum at 80 °C. After each consecutive run, the above steps were repeated with the mixture. According to the optimal experimental conditions, recovering [CPL][2MSA] was reused for ten runs without adding new substances.

### *3.6. COSMO-RS*

The thermophysical properties of the molecular surface charge densities ($\sigma$) of ILs were calculated by COSMO-RS model. The first step is that chemical structures of ILs were optimized using TmoleX (Molex, Lisle, IL, USA) (Version 4.5 N) software at the density functional theory (DFT) with empirical dispersion correction level, apply with parameterization BP_TZVPD_FINE_20.ctd). All energy calculations and the estimated molecular surface charge density values in the ILs have been stored as .cosmo files into COSMOthermX (version 19.0.0 COSMOlogic GmbH & Co. KG, Leverkusen, Germany) for further thermophysical property calculations.

### *3.7. DFT Calculation*

The Gaussian View 5.0 (Gaussian, New York, NY, USA) was used to draw all the molecular structure. All the compounds of geometrical optimizations were performed in gas phase by using the DFT 6-311G + (d, p) basis set. The molecular orbitals known as

HOMO & LUMO and the Molecular Electrostatic Potential were performed employing the same basis sets.

## 4. Conclusions

In this work, [CPL][2MSA], as a novel Brønsted acidic ionic liquids, has been synthesized by one step solvent-free process. It should be pointed out that [CPL][2MSA] exhibits the high conversion (100%) and selectivity (95%) without other any co-catalysts or metals. Based on the results of thermogravimetric analysis (TGA) and differential scanning calorimetry (DSC), the decomposition and glass transition temperatures are gradually increase with the increase of MSA mole content. The path of the reaction is further verified by in-situ FT-Raman spectra and COSMO-RS conclusion, and the result shows that the H-bond is the dominant mode in this catalytic system. Interestingly, the process of converting CHO to CPL follows the first order reaction kinetics, and the required activation energy for this process is calculated (29.8 KJ $mol^{-1}$). In conclusion, this work not only provides a new environmentally friendly and low-cost catalyst ([CPL][2MSA]), but also develops a feasible technical route for the CPL production process.

**Supplementary Materials:** The following supporting information can be downloaded at: https://www.mdpi.com/article/10.3390/catal13060978/s1. Figure S1. NMR spectrum of [CPL][2MSA]; Figure S2. The electrochemical spectra of different samples; Figure S3. Predicted results for capacity of CHO in [CPL][2MSA] by COSMO-RS; Figure S4. Excess enthalpies of CHO in [CPL][2MSA] at *T* = 363.15 K.

**Author Contributions:** C.R. and Z.W.; writing, conceptualization, formal analysis, investigation, and data curation; Q.G., J.L., S.J., Q.H. and Y.Y.; resources, methodology; J.Z. and Y.W.; validation; Z.L., Y.H. and X.G.; conceptualization, project administration, funding acquisition, resources. All authors have read and agreed to the published version of the manuscript.

**Funding:** The authors gratefully acknowledge financial support from the National Natural Science Foundation of China (grant numbers 21776300, 21890763, and 22078355) and the Science and Technology Department of Qinghai Province (grant number 2022-GX-152).

**Data Availability Statement:** The authors confirm that the data supporting the findings of this study are available within the article and its supplementary materials. Raw data that support the findings of this study are available from corresponding authors, upon reasonable request.

**Conflicts of Interest:** The authors declare no conflict of interest.

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
