# Peer review of "Novel Brønsted Acidic Ionic Liquids as High Efficiency Catalysts for Liquid-Phase Beckmann Rearrangement"

_catalysts, doi:10.3390/catal13060978_

Round 1

Reviewer 1 Report

The manuscript describes the use of Bronsted Acidic Ionic liquids for Liquid-Phase Beckmann rearrangement. No doubt this is an important area and much room for improvement is a reality. Regarding this manuscript, although it is important, several issues require the authors' attention. However, the caprolactam-based ionic liquids were previously used in the Beckmann rearrangement by Deng’s group (Green Chem., 2006, 8, 296-300). Later, caprolactam hydrosulfate, caprolactam p-toluenesulfonate, caprolactam benzenesulfonate, etc were also synthesized and used in different reactions (Synth. Commun. 2008, 38, 537–545, J. Mol. Catal. A Chem. 2014, 383, 101–105, Catalysts 2017, 7, 102; doi:10.3390/catal7040102 etc.) The authors used methane sulphonic acid with caprolactam and checked the catalytic activity in the Beckmann rearrangement. The obtained results with this catalytic system are in better conversion and selectivity than the previous work (reference 23).

1. Does authors reported the mass spectra of the novel synthesized catalysts? Provide the HRMS spectra of newly synthesized ionic liquids.

2. In the Table 3. Results of the liquid phase Beckmann rearrangement of cyclohexanone oxime to caprolactam. What is the superscripts a and b of entries 1, 5, 6, and 7 stand for? Add footnotes related to this.

3. Why not the ionic liquids are shown as ion pairs as in the other literature reports?

4.  Why author chooses only methane sulphonic acid to synthesize catalysts? Does authors check the effect of [CPL][hydrogen sulfate], [CPL][p-toluenesulfonate] or any other sulphonic acid in this reaction? If not, then authors can check the effect of these synthesized catalysts in the Beckmann rearrangement?

5. Author should show the substrate scope by using different substrates for the Beckmann rearrangement.

6- This manuscript has several typos and grammar issues. These problems should be corrected. Some of them are given below.

·         In the results and discussion parts, in the 9th line bule shifted changed to blue shifted

·         In the results and discussion parts, in the 24th line MAS corrected to MSA

·         A little explanation of Table 3 should be given in text also.

·         Two tables are given numbering as table 3. So numbering of tables should be corrected in tables as well as in text.

·         In page no 9, the Ist paragraph CPL (g) is not in figure 7. Corrected CPL (g) to CPL (c)

·         In the materials and methods section, 3.2.1 CHCl3 is replaced by CDCl3

In the materials and methods section, section 3.5, then then word is repeating. Delete one word.

After major revision manuscript may be accepted for publication.

This manuscript has several typos and grammar issues. These problems should be corrected. Some of them are given below.

·         In the results and discussion parts, in the 9th line bule shifted changed to blue shifted

·         In the results and discussion parts, in the 24th line MAS corrected to MSA

·         A little explanation of Table 3 should be given in text also.

·         Two tables are given numbering as table 3. So numbering of tables should be corrected in tables as well as in text.

·         In page no 9, the Ist paragraph CPL (g) is not in figure 7. Corrected CPL (g) to CPL (c)

·         In the materials and methods section, 3.2.1 CHCl3 is replaced by CDCl3

In the materials and methods section, section 3.5, then then word is repeating. Delete one word.

Author Response

Manuscript ID: GCENG-D-22-00089

Title: Novel Brønsted Acidic Ionic Liquids as High Efficiency Catalysts for Liquid-Phase Beckmann rearrangement

Author(s): Chunxiao Ren a,b,1, Zhiyuan Wang a,1, Qingwei Gao a, Jisheng Li a, Siqi Jiang c, Qizhong Huang a, Ye Yang a, Jiahui Zhang a, Yajing Wang a, Yufeng Hu*, a, Zhichang Liu*, a, Xin Guo*, a

Dear Editor

Thank you very much for your kind E-mail and your kind help. We revised the present MS in line with the editorial comments and the valuable comments of the reviewers. Therefore, some necessary changes have been made as follows (the changes have been highlighted in yellow).

Response to Reviewer 1:

The manuscript describes the use of Bronsted Acidic Ionic liquids for Liquid-Phase Beckmann rearrangement. No doubt this is an important area and much room for improvement is a reality. Regarding this manuscript, although it is important, several issues require the authors' attention. However, the caprolactam-based ionic liquids were previously used in the Beckmann rearrangement by Deng’s group (Green Chem., 2006, 8, 296-300). Later, caprolactam hydrosulfate, caprolactam p-toluenesulfonate, caprolactam benzenesulfonate, etc were also synthesized and used in different reactions (Synth. Commun. 2008, 38, 537–545, J. Mol. Catal. A Chem. 2014, 383, 101–105, Catalysts 2017, 7, 102; doi:10.3390/catal7040102 etc.) The authors used methane sulphonic acid with caprolactam and checked the catalytic activity in the Beckmann rearrangement. The obtained results with this catalytic system are in better conversion and selectivity than the previous work (reference 23).

  1. Does authors reported the mass spectra of the novel synthesized catalysts? Provide the HRMS spectra of newly synthesized ionic liquids.

REPLY: Thanks for this valuable comment.

MODIFICATION: We have tested the 1H NMR and 13C NMR (Fig. S1a, b) of the novel synthesized catalysts for [CPL][2MSA] in Supporting Information.

  1. In the Table 3. Results of the liquid phase Beckmann rearrangement of cyclohexanone oxime to caprolactam. What is the superscripts a and b of entries 1, 5, 6, and 7 stand for? Add footnotes related to this.

REPLY: Thanks for this valuable comment.

MODIFICATION: According to the suggestion. We have deleted the incorrect expression (a and b) of entries 1, 5, 6, and 7 in Table 4 of the revised MS. In addition, due to a misnumbering of the table, Table 3 should be called Table 4, which has been amended accordingly in the revised MS. The revised parts are marked in the yellow color in the revised manuscript.x

  1. Why not the ionic liquids are shown as ion pairs as in the other literature reports?

REPLY: Thanks for this valuable comment.

MODIFICATION: Usually, ionic liquids are synthesized by ion pairs. Moreover, in some cases, various amounts of methanesulfonic acid, as anions, have been introduced in ionic liquids to enhance acidity, and then improved the catalytic effect. In this work, a novel Brønsted acidic ionic liquids based on caprolactam bases and methanesulfonic acid has been developed. We have added the mentioned references and some contents in Page 3. The following references “Green Chem., 2014, 16, 3463–3471, Green Chem. 2002, 4, 134–138.” have been cited in Introduction of revised MS as 27, 29 reference numbers. The revised parts are marked in the yellow color in the revised manuscript.

  1. Why author chooses only methane sulphonic acid to synthesize catalysts? Does authors check the effect of [CPL][hydrogen sulfate], [CPL][p-toluenesulfonate] or any other sulphonic acid in this reaction? If not, then authors can check the effect of these synthesized catalysts in the Beckmann rearrangement?

REPLY: Thanks for this valuable comment.

MODIFICATION: According to the suggestion. Under the optimal sample synthesis method, [CPL][2Cl], [CPL][2BF4], [CPL][2CSA], [CPL][2HSO4]( [CPL][hydrogen sulfate]), [CPL][2TsOH]([CPL][p-toluenesulfonate]), [CPL][2TfOH] have been synthesized, and then tested the effect of these synthesized catalysts in the Beckmann rearrangement. Compared with [CPL][2MSA], the results showed that the lower selectivity (CPL) was obtained for the above-synthesized ILs catalysts. The experimental results have been added in Table 4 (entries 8–13) of the revised MS. In addition, the experimental results can also confirm the superiority of the reported [CPL][2MSA] in this work. Moreover, we have also added the mentioned references by the reviewer in comment “Synth. Commun. 2008, 38, 537–545, J. Mol. Catal. A Chem. 2014, 383, 101–105, Catalysts 2017, 7, 102.” as 54, 55, 56 reference numbers of the revised MS. The revised parts are marked in the yellow color in the revised manuscript.

  1. Author should show the substrate scope by using different substrates for the Beckmann rearrangement.

REPLY: Thanks for this valuable comment.

MODIFICATION: According to the suggestion. We have explored the substrate scope of Beckmann rearrangement catalyzed by [CPL][2MSA], various ketoximes substrates were examined. In Table 6 (Entries 1–4), aromatic ketoximes show the higher yields 92.7%–98.4% and cyclododecanone oxime also exhibits a comparative yield (94.9%) under the same experimental conditions. This again evidences the high catalytic activity of [CPL][2MSA] for Beckmann reactions. We have added some contents in Page 12. The revised parts are marked in the yellow color in the revised manuscript.

  1. This manuscript has several typos and grammar issues. These problems should be corrected. Some of them are given below.

In the results and discussion parts, in the 9th line bule shifted changed to blue shifted

REPLY: Thanks for this valuable comment.

MODIFICATION: According to the suggestion. “bule shifted” has been replaced by “blue shifted” in the 9th line of the revised MS. The revised parts are marked in the yellow color in the revised MS.

In the results and discussion parts, in the 24th line MAS corrected to MSA

REPLY: Thanks for this valuable comment.

MODIFICATION: According to the suggestion. In the results and discussion parts, “MAS” has been replaced by “MSA” in the 24th line of the revised MS. The revised parts are marked in the yellow color in the revised MS.

A little explanation of Table 3 should be given in text also.

REPLY: Thanks for this valuable comment.

MODIFICATION: According to the suggestion. We have added the relevant explanation of Table 3 in Page 5 of the revised MS. The revised parts are marked in the yellow color in the revised MS.

Two tables are given numbering as table 3. So numbering of tables should be corrected in tables as well as in text.

REPLY: Thanks for this valuable comment.

MODIFICATION: According to the suggestion. Incorrect order of Tables has been revised in the revised MS. The revised parts are marked in the yellow color in the revised manuscript.

In page no 9, the Ist paragraph CPL (g) is not in figure 7. Corrected CPL (g) to CPL (c).

REPLY: Thanks for this valuable comment.

MODIFICATION: According to the suggestion. “CPL (g)” has been replaced by “CPL (c)” in Page 9 of the revised MS. The revised parts are marked in the yellow color in the revised MS.

In the materials and methods section, 3.2.1 CHCl3 is replaced by CDCl3.

REPLY: Thanks for this valuable comment.

MODIFICATION: According to the suggestion. In the materials and methods section, “CHCl3” has been replaced by “CDCl3” in 3.2.1 section of the revised MS. The revised parts are marked in the yellow color in the revised MS.

In the materials and methods section, section 3.5, then then word is repeating. Delete one word.

REPLY: Thanks for this valuable comment.

MODIFICATION: According to the suggestion. In the materials and methods section, section 3.5, the repeated word (then) has been deleted in the revised MS. The revised parts are marked in the yellow color in the revised MS.

Reviewer 2 Report

The work proposed by Hu, Liu and Guo treats on the synthesis of a caprolactam based IL and its application for a Beckmann rearrangement. The overall question that emerges concerning this work is: why one should first synthetize and then use a caprolactam-based IL to synthetize finally the caprolactam? How it is possible to discriminate between the caprolactam from the IL and from the reaction product? I do not understand the aim of the work, which finally results foggy and plenty of extra experiments to calculate and demonstrate things that are already known (ACS Sustainable Chem. Eng. 2022, 10, 41, 13568–13575and not related to the catalysis per se. The main text results confusing and often complicated to follow. Periods are sometimes too long or words (and verbs) are missing. There is a lack of a solid and potentially reproducible experimental procedure and the conclusions are overselling this transformation (which is not catalytic at all, albeit this is the claim of the work!) To conclude, in my opinion, the scientific soundness of this article is not suitable for a publication in Catalysts nor other journals treating the field.  

See comments and suggestions to the authors section

Author Response

Manuscript ID: GCENG-D-22-00089

Title: Novel Brønsted Acidic Ionic Liquids as High Efficiency Catalysts for Liquid-Phase Beckmann rearrangement

Author(s): Chunxiao Ren a,b,1, Zhiyuan Wang a,1, Qingwei Gao a, Jisheng Li a, Siqi Jiang c, Qizhong Huang a, Ye Yang a, Jiahui Zhang a, Yajing Wang a, Yufeng Hu*, a, Zhichang Liu*, a, Xin Guo*, a

Dear Editor

Thank you very much for your kind E-mail and your kind help. We revised the present MS in line with the editorial comments and the valuable comments of the reviewers. Therefore, some necessary changes have been made as follows (the changes have been highlighted in yellow).

Response to Reviewer 2:

The work proposed by Hu, Liu and Guo treats on the synthesis of a caprolactam based IL and its application for a Beckmann rearrangement. The overall question that emerges concerning this work is: why one should first synthetize and then use a caprolactam-based IL to synthetize finally the caprolactam? How it is possible to discriminate between the caprolactam from the IL and from the reaction product? I do not understand the aim of the work, which finally results foggy and plenty of extra experiments to calculate and demonstrate things that are already known (ACS Sustainable Chem. Eng. 2022, 10, 41, 13568–13575) and not related to the catalysis perse. The main text results confusing and often complicated to follow. Periods are sometimes too long or words (and verbs) are missing. There is a lack of a solid and potentially reproducible experimental procedure and the conclusions are overselling this transformation (which is not catalytic at all, albeit this is the claim of the work!) To conclude, in my opinion, the scientific soundness of this article is not suitable for a publication in Catalysts nor other journals treating the field.

  1. Why one should first synthetize and then use a caprolactam-based IL to synthetize finally the caprolactam?

REPLY: Thanks for this valuable comment.

MODIFICATION: In this work, the rearrangement reaction product as one component of ILs used, can be conjectured that there would be a dynamic exchange between caprolactam ILs and the produced caprolactam during the rearrangement reaction, and the strong chemical combination between caprolactam and acidic catalyst would be largely avoided. In addition, the product would be easily separated from ILs by using organic solvent extraction. We have added the relevant explanation in Page 3 of the revised MS. The revised parts are marked in the yellow color in the revised MS.

  1. How it is possible to discriminate between the caprolactam from the IL and from the reaction product?

REPLY: Thanks for this valuable comment.

MODIFICATION: In this work, based on the COSMO-RS model (Fig. 6), it can be seen that caprolactam has both the ability of hydrogen bond acceptor and hydrogen bond donor, indicating that caprolactam, as a cation, easily forms hydrogen bond with anion. According to the results of physicochemical properties (Fig. 2), the decomposition temperatures (Tg) of caprolactam-based ILs are significantly higher than caprolactam. It also verifies that caprolactam-based ILs are composed between anions and cations by the connection with the hydrogen bond. In addition, the reaction product (caprolactam) is a solid and caprolactam-based ILs is a liquid. After the rection, the product was washed three times through organic solvent (acetone), and then dried in an oven at 80 °C for 12 h. Finally, the organic solvent (acetone) was removed by using a rotary evaporator, recovering [CPL][2MSA]. We have added the relevant explanation of in Page 15 of the revised MS. The revised parts are marked in the yellow color in the revised MS.

  1. I do not understand the aim of the work, which finally results foggy and plenty of extra experiments to calculate and demonstrate things that are already known (ACS Sustainable Chem. Eng. 2022, 10, 41, 13568–13575) and not related to the catalysis perse.

REPLY: Thanks for this valuable comment.

MODIFICATION: Compared with reported literature, this article successfully developed a low-cost and environmentally friendly catalyst with a special structural design. The catalytic selectivity (CPL) and required activation energy of the novel catalyst are significantly lower than most reported literature. In addition, COSMO-RS model as a powerful thermodynamic tool, it only requires the structural information for to predict the solubility and other thermodynamic properties. In this work, it confirmes through COSMO-RS model that CPL and MSA not only exhibit the ability to interact the H-bond donor, but also the H-bond acceptor. This provides effective evidence for explaining the Beckmann rearrangement mechanism. We also predict the solubility of CHO in [CPL][2MSA] at the different temperature, the result shows that the solubility of CHO gradually increases from 25 ℃ to 90 ℃. If the temperature is above 90 ℃, the solubility of CHO sharply descends, this result is in accordance with the experimental results, and then it is beneficial for promoting the progress of the experiment based on the predicted results. The above results have not been reported so far. Therefore, this work still has reference value for promoting caprolactam production industry. As far as we know, the concentrated sulfuric acid or oleum is mainly used as the classical Beckmann rearrangement catalyst to catalyze the conversion of cyclohexanone oxime to caprolactam in large-scale industrial production. However, the process shows disadvantages of heavy corrosion and a large amount of by-product ammonium sulphate. As such, there has been considerable interest in designing efficient catalysts for the industrial Beckmann rearrangement. Therefore, this work is closely related to catalysis. It can be found in many references, such as “, Nature 2005, 437, 1243–1244, Catal. Lett. 2013, 143, 193–199.”.

  1. Periods are sometimes too long or words (and verbs) are missing. There is a lack of a solid and potentially reproducible experimental procedure and the conclusions are overselling this transformation (which is not catalytic at all, albeit this is the claim of the work!)

REPLY: Thanks for this valuable comment.

MODIFICATION: According to the suggestion. We have carefully checked the English expressions of the whole manuscript and corrected some mistakes. In addition, we have revised the detailed process of the repetitive experiment and the conclusions in Page 15, 16 of the revised MS. The revised parts are marked in the yellow color in the revised MS.

Reviewer 3 Report

The research described in this manuscript is interesting, although I am not convinced that it is as major of a step forward as is presented.  There are an enormous number of ways to conduct Beckmann rearrangements and, while the summary presented is okay, it is certainly not comprehensive.  It also fails to fully compare the range of considerations that would go into deciding which conditions to use (yield, time, temperature, cost, method of isolation, purity, environmental impact of the entire sequence) and simply picks a few examples to support the conclusions that you, the authors, want.  As such, it is not objective, but the review is simply used as justification for your conditions.  Indeed, in some cases you are contradictory.  You talk about the sue of organic solvents as being bad, but then you use organic solvents to extract the product (neither acetone nor diethyl ether are acceptable industrially).  In terms of why CPL 2MSA is superior, you provide very limited explanation.  That may be understandable (the reason may be unclear), but I do wonder if you have looked at how hygroscopic the compounds are (CPL and the three adducts with varying amounts of MSA).  Could it be that CPL 2MSA is better able to sequester the water formed during the reaction and thus minimize hydrolysis as a competing reaction?  Clearly the CPL/MSA adducts how a much higher retention of water as measured in your Karl-Fischer work.  I presume that these measurements were made after drying, so you may want to look at water equilibrated samples, too.

I am uncertain if the lengthy set of equations associated with the activation energy determination are really useful.  I am not familiar with this being a typical part of such manuscripts and am honestly not certain where some of these equations even came from (despite the fact that I do kinetic work in my own research).  Either better context, referencing, or maybe deletion should be considered.

In looking at the experimental, it seems that all of the CPL/MAS adducts gave the same NMR spectra.  What does that mean?  Your reported drying conditions would not remove excess MSA, so I am left wondering if the different ratios are really stable or simply reflect the initial molar ratio combined.  I know that the TGA does not indicate a mass loss, so I am uncertain how to reconcile this data as well as the clear reported reaction differences.  This aspect should be clarified.

Finally, while you mention using other acids with CPL, it would be nice in the supporting information to give a clearer indication of what was explored and what results were obtained.  This will help others to potentially use or even build on this research work.

This manuscript needs a lot of work for the English to be even just adequate.  While I could eventually follow most of the text, it was not easy and multiple sentences on every page left me confused as to their intent.

Author Response

Manuscript ID: GCENG-D-22-00089

Title: Novel Brønsted Acidic Ionic Liquids as High Efficiency Catalysts for Liquid-Phase Beckmann rearrangement

Author(s): Chunxiao Ren a,b,1, Zhiyuan Wang a,1, Qingwei Gao a, Jisheng Li a, Siqi Jiang c, Qizhong Huang a, Ye Yang a, Jiahui Zhang a, Yajing Wang a, Yufeng Hu*, a, Zhichang Liu*, a, Xin Guo*, a

Dear Editor

Thank you very much for your kind E-mail and your kind help. We revised the present MS in line with the editorial comments and the valuable comments of the reviewers. Therefore, some necessary changes have been made as follows (the changes have been highlighted in yellow).

Response to Reviewer 3:

The research described in this manuscript is interesting, although I am not convinced that it is as major of a step forward as is presented. There are an enormous number of ways to conduct Beckmann rearrangements and, while the summary presented is okay, it is certainly not comprehensive. It also fails to fully compare the range of considerations that would go into deciding which conditions to use (yield, time, temperature, cost, method of isolation, purity, environmental impact of the entire sequence) and simply picks a few examples to support the conclusions that you, the authors, want.

REPLY: Thanks for this valuable comment.

MODIFICATION: In this work, compared with imidazolium, caprolactam are cheaper, more environmentally benign, and available on an industrial scale, which would be attractive in industry if incorporated into ILs as catalyst and medium for Beckmann rearrangement. In addition, we have investigated the effects of yield, time, temperature and molar ratio (reactant: catalyst), and the relevant experimental results have been shown in the Fig. 5(a, b, c) of the revised MS. The type of catalyst, the ratio of reactant and catalyst, and the cyclic stability of the catalyst have been further investigated in Table 4 and Fig. 5(d) of the revised MS. According to literature research, most of ILs are separated by extraction. This article used acetone as an extractant because it can extract the ionic liquid after the reaction, and recovered the ionic liquid for recycling experiments. It can be found in many references, such as “Green Chem., 2006, 8, 296–300.”. This paper focuses on the catalysis and mechanism of liquid phase Beckmann rearrangement. Therefore, no in-depth consideration was investigated for extractants, we will further consider the relevant studies in our future work. For environmental impact of the entire sequence, ionic liquids (ILs) as environmentally friendly reaction media and catalysts have been widely recognized and accepted. ILs also exhibit several attractive properties such as negligible vapor pressure, non-flammable, non-toxic, high thermal and chemical stability, and its use as a catalyst for specific reactions. A slight modification in cation and/or anion structure results in an enormous range of potential value for ILs. It has been demonstrated that the anion of IL plays a dominant role in the catalysis. The relevant literature has been reported, such as “Green Chem., 2022, 24,4140–4152, Chem. Rev. 2007, 107, 2615−2665.”.

As such, it is not objective, but the review is simply used as justification for your conditions. Indeed, in some cases you are contradictory. You talk about the sue of organic solvents as being bad, but then you use organic solvents to extract the product (neither acetone nor diethyl ether are acceptable industrially).

REPLY: Thanks for this valuable comment.

MODIFICATION: According to literature research, most of ILs are separated by extraction. This article used acetone as an extractant because it can extract the ionic liquid after the reaction, and recovered the ionic liquid for recycling experiments. It can be found in many references, such as “Green Chem., 2006, 8, 296–300.”. This paper focuses on the catalysis and mechanism of liquid phase Beckmann rearrangement. Therefore, no in-depth consideration was investigated for extractants, we will further consider more environmentally friendly extractants for the separation of ionic liquids and products in our future work. We have added the relevant explanation of in Page 15 of the revised MS. The revised parts are marked in the yellow color in the revised MS.

In terms of why CPL 2MSA is superior, you provide very limited explanation. That may be understandable (the reason may be unclear), but I do wonder if you have looked at how hygroscopic the compounds are (CPL and the three adducts with varying amounts of MSA). Could it be that CPL 2MSA is better able to sequester the water formed during the reaction and thus minimize hydrolysis as a competing reaction? Clearly the CPL/MSA adducts how a much higher retention of water as measured in your Karl-Fischer work. I presume that these measurements were made after drying, so you may want to look at water equilibrated samples, too.

REPLY: Thanks for this valuable comment.

MODIFICATION: According to the suggestion. We have tested the content of water after the reaction. Compared with the dried ILs before the reaction, the content of water of [CPL][2MSA] and [CPL][3MSA] get higher with the increased content of MAS. The results have been added in Table 2 (entry 9). I presume that [CPL][2MSA], [CPL][3MSA], containing an excess of MSA, can connect with water molecules. Finally, it leads to increase the water content of samples. If the content of MSA is too high, resulting in high acidity of the ILs and then decreased in the selectivity of the catalytic system. As the reviewer guess, the optimal [CPL][2MSA] used in this article is better able to sequester the water formed during the reaction and thus minimize hydrolysis as a competing reaction. We have added the relevant explanation of in Page 8 of the revised MS. The revised parts are marked in the yellow color in the revised MS.

I am uncertain if the lengthy set of equations associated with the activation energy determination are really useful. I am not familiar with this being a typical part of such manuscripts and am honestly not certain where some of these equations even came from (despite the fact that I do kinetic work in my own research). Either better context, referencing, or maybe deletion should be considered.

REPLY: Thanks for this valuable comment.

MODIFICATION: In our work, [CPL][2MSA] with the advantages of high yield and low activation energy, makes it possible to carry out the reaction at mild conditions, i.e., 90 °C for 2 h. In Table 5, it also can be seen that activation energy required for [CPL] [2MSA] is lower than other reported. The conversion of cyclohexanone oxime to caprolactam has been confirmed for the first-order reaction kinetics. It can be found in many references, such as “AIChE J. 2018, 64, 571-577, Chem. Eng. Sci. 2022, 253 117519”. We have added the relevant explanation and the relevant references in Page 12 of the revised MS. The revised parts are marked in the yellow color in the revised MS.

In looking at the experimental, it seems that all of the CPL/MAS adducts gave the same NMR spectra. What does that mean? Your reported drying conditions would not remove excess MSA, so I am left wondering if the different ratios are really stable or simply reflect the initial molar ratio combined. I know that the TGA does not indicate a mass loss, so I am uncertain how to reconcile this data as well as the clear reported reaction differences. This aspect should be clarified.

REPLY: Thanks for this valuable comment.

MODIFICATION: This work only tested the Nuclear Magnetic Resonance for the optimal samples as shown in Fig. S1(a, b). In addition, the FT-IR spectra of [CPL][XMSA] have been shown in Fig. 1a, with the increase of MSA molar ratios, the S–OH vibration are both blue shifted for [CPL][2MSA] and [CPL][3MSA] from 842.2 cm–1 to 942.5 cm–1. Therefore, the stronger H-bond are connected between MSA and [CPL][MSA]. Based on the COSMO-RS model (Fig. 6), CPL and MSA not only exhibit the ability to interact the H-bond donor, but also the H-bond acceptor, it can further verify the stronger combine between CPL and MSA. The results of the conductivities and electrode potentials also confirmed the presence of the hydrogen bond. Due to the presence of the strong H-bond, the introducing different ratios for MAS on CPL are really stable. We have added the relevant explanation in Page 4 of the revised MS. The revised parts are marked in the yellow color in the revised MS.

Finally, while you mention using other acids with CPL, it would be nice in the supporting information to give a clearer indication of what was explored and what results were obtained. This will help others to potentially use or even build on this research work.

REPLY: Thanks for this valuable comment.

MODIFICATION: This work is based on the optimal experimental conditions for the [CPL][2MSA] to compare with the other type of catalysts. We also further investigated the catalytic effect of CPL with other types of acids and then the relevant expression and experimental results have been added in 3.3 section and Table 4 of the revised MS. The revised parts are marked in the yellow color in the revised manuscript.

This manuscript needs a lot of work for the English to be even just adequate. While I could eventually follow most of the text, it was not easy and multiple sentences on every page left me confused as to their intent.

REPLY: Thanks for this valuable comment.

MODIFICATION: According to the suggestion. We have carefully checked the English expressions of the whole manuscript and corrected some mistakes. The revised parts are marked in the yellow color in the revised MS.
